Subject Area:
cellular biology/developmental biology/genetics

Keywords:
oesophageal carcinoma, gastric carcinoma, neoadjuvant treatment, patient-derived organoid, response prediction

Authors for correspondence:
Hans Clevers
e-mail: h.clevers@hubrecht.eu
Michiel F. G. de Maat
e-mail: m.f.g.demaat-3@umcutrecht.nl

# The potential and challenges of patient-derived organoids in guiding the multimodality treatment of upper gastrointestinal malignancies

Georg A. Busslinger[1], Fianne Lissendorp[2], Ingrid A. Franken[1], Richard van Hillegersberg[2], Jelle P. Ruurda[2], Hans Clevers[1] and Michiel F. G. de Maat[2]

[1]Royal Netherlands Academy of Arts and Sciences, Oncode Institute and Hubrecht Institute, 3584CT Utrecht, The Netherlands
[2]Department of Surgery, University Medical Center Utrecht, PO Box 85500, 3508GA Utrecht, The Netherlands

GAB, 0000-0002-7851-9895; HC, 0000-0002-3077-5582

The incidence of adenocarcinoma at the gastrooesophageal junction increased over the last years. Curative treatment for patients with upper gastrointestinal (UGI) malignancies, such as oesophageal and gastric tumours, is challenging and requires a multidisciplinary approach. Radical surgical resection with complete lymphadenectomy is the cornerstone of UGI cancer treatment. Combined with peri-operative treatment (i.e. by applying CROSS, EOX or FLOT regimen), the survival is even better than with surgery alone. However, peri-operative treatment is not effective in all patients, and the most effective strategy is a topic of active debate, as is reflected by varying treatment guidelines between countries. UGI cancers are (epi)genetically highly heterogeneous. It is thus not likely that a uniform treatment will benefit all patients equally well. Over recent years, patient-derived organoids (PDOs) gained more and more interest as an *in vitro* prediction model that may assist as a diagnostic tool in the future to select and eventually optimize the best peri-operative treatments for each patient. PDOs can be derived from endoscopic tumour biopsies, which maintain heterogeneity in culture. They can be rapidly established and expanded in a relatively short time for *in vitro* drug screening experiments. This review summarizes the clinical and molecular aspects of oesophageal and gastric tumours, as well as the current progress and remaining challenges in the use of PDOs for drug and radiation screens.

## 1. Introduction

Malignant tumour transformations in the upper gastrointestinal (UGI) tract are preferentially occurring within the oesophagus and stomach, accounting for respectively 4.9% and 8.8% of the total cancer deaths worldwide [1]. Although the incidence of gastric cancer is decreasing, the number of people diagnosed with oesophageal cancer has increased in recent years. This is mostly attributed to a rise in gastrooesophageal junction (GOJ) tumours, especially in the Western world [2]. In addition to forming an anatomical continuum, oesophageal and gastric cancer show similar heterogeneity in both clinical response to multimodality treatment and molecular characteristics. This review addresses and integrates clinical and molecular perspectives with the ultimate aim to outline the potential applications and challenges in using patient-derived organoids (PDOs) as an *in vitro* model to recapitulate the *in vivo* tumour behaviour and response to multimodality treatment.

## 2. Clinical perspective

Although treatment with curative intent of UGI cancers is feasible in some cases, patients are commonly diagnosed at an advanced stage with local growth (cN+ or cT3-4) into surrounding tissues, seeding to the peritoneal cavity as a specific feature of gastric cancer, or early systemic dissemination (cM+). Unfortunately, once metastasized, curative options are limited since effective systemic drugs are scarce. In locally confined tumours, radical surgical resection with complete lymphadenectomy is the cornerstone of UGI cancer treatment. Local control is challenging due to the vital anatomical structures surrounding the oesophagus and stomach, such as the aorta, trachea and pancreas. Therefore, the relation of a cT3-4 tumour to the surrounding tissues necessitates effective downstaging to obtain microscopically free resection margins during surgery. These complications of UGI cancers impose a need for (neo)adjuvant treatment to enable curative treatment for advanced disease stages. There is no consensus on whether optimal (neo)adjuvant treatment should focus on locoregional or systemic control, as is illustrated by the current treatment guidelines that differ between countries. For instance, when a patient is diagnosed with a distal oesophageal adenocarcinoma (OAC) in the United Kingdom, systemic triplet drugs are advised, aiming to both downsize locally and eradicate tumour cells systemically (according to the MAGIC trial: epirubicin and capecitabine combined with cisplatin (ECX) or oxaliplatin (EOX) in systemic dose [3]). In contrast, in The Netherlands, the same patient would receive neoadjuvant chemoradiation focusing on local control and nodal sterilization (according to the CROSS schedule: 23 fractions of 1.8 Gy with low radio-sensitizing doses of paclitaxel and cisplatin weekly as chemosensitizers [4]). There is no consensus on the best approach, although response rates of both regimens are comparable. Improved response rates are obtained in gastric and GOJ adenocarcinoma by triplet regimens such as FLOT (combining docetaxel, oxaliplatin, leucovorin and 5-fluouracil [5]), offering hope that combined effective systemic and locoregional control is possible. The current regimens are not effective in all patients. One patient might benefit from EOX or FLOT, another from CROSS and another from direct surgery. To improve insights into the best approach for treating oesophageal cancer, a large multicentre randomized international trial (NeoAgis [5,6]) is currently including patients with distal OACs to receive either systemic therapy (EOX or FLOT) or chemoradiation (CROSS).

Different treatment strategies also exist for oesophageal squamous cell carcinoma (OSCC), which is generally found in the upper and middle third of the oesophagus. OSCC responds to chemoradiation two times better than OAC [4]. In The Netherlands, potentially resectable OSCCs are treated, similar to OACs, with CROSS regimen followed by surgery, whereas in France, a patient with OSCC is scheduled for definitive chemoradiation and will be only operated on in case of a tumour regrowth or detection of residual tumour tissue. On the one hand, it would be beneficial to avoid surgical resection if there is a high chance of a complete response. On the other hand, omitting surgery also exposes a subgroup of patients to interval metastases or salvage esophagectomy, which comes with a higher peri-operative morbidity than direct surgery after chemoradiation [7].

In all (neo)adjuvant treatment regimens, only a minority of the patients show a (near)complete response, accounting for 29% in OAC patients to CROSS and 16% of GOJ or gastric adenocarcinoma (GAC) to FLOT [8]. The individual differences in treatment response and the resulting inadequate treatment of a subset of patients highlight the need to progress from a uniform treatment towards an individualized treatment. As will be discussed in the next sections, the differences in treatment response may result from the genetic heterogeneity in UGI cancers. However, efforts to explain and predict whether an individual will respond to treatment have to date not yielded clinically applicable biomarkers. An attractive alternative is to assess the tumour response of patients by an individualized assay *a priori* to its actual administration. *In vitro* organoid cultures hold the potential to recapitulate *in vivo* tumour behaviour and ultimately tailor the neoadjuvant treatment strategy for each patient [9].

## 3. Molecular heterogeneity of UGI cancers

Over the last few years, the genomic tumour landscape was extensively studied with high-throughput sequencing methods such as whole-genome/exome sequencing, DNA methylation-based profiling, mRNA sequencing and analysis of somatic copy-number alterations for either oesophageal tumours [10,11], gastric carcinoma [12,13] or comparing tumours spanning the entire gastrointestinal tract (GI) [14]. Comparison of all gastrointestinal adenocarcinomas (GIACs) to other cancer types, including lung and breast, indicate that GIACs constitute a unique entity with specific mutations (e.g. in *ATM*), amplifications (e.g. in *GATA4/6*, *EGFR*, *CD44*, *FGFR1*, *IGF2*) and a higher hypermethylation frequency [14].

Among the oesophageal cancers, there is a clear molecular distinction between OSCC and OAC. Although both subtypes carry *TP53* mutations and inactivation of *CDKN2A*, the additional mutations and chromosomal instabilities diverge substantially [11]. OSCC have a higher resemblance to head and neck squamous cell carcinoma [11], whereas OAC and GAC cluster together [11,14]. An independent study focusing on 551 OACs [10] expanded the list of potential driver mutations, which are located in either coding genes or non-coding elements, such as promoters and enhancers. Similar to other studies, massive chromosomal amplifications and deletions are observed, but interestingly only a few of them are predicted to cause significant gene expression changes. These include amplifications in *ERBB2*, *KRAS* and *SMAD4*, or deletions in *ARID1A* and *CDH11*.

Historically, gastric cancers are subdivided based on histopathological analysis into intestinal, diffuse and mixed types according to the Lauren classification [15]. Recent sequencing efforts divide gastric cancers into four molecular subgroups: Epstein–Barr virus (EBV)-positive, microsatellite instable (MSI), chromosomal instable (CIN) and genomic stable (GS) tumours [12–14]. The subgroups do not harbour apparent regional specificity, except for the EBV-positive cancers, or obvious correlation to the Lauren classification. Although not exclusive, the diffuse-type cancer is enriched in GS subgroup (73% [12] or 66% [14]). GS cancers are characterized by low chromosomal aberrations, low mutation frequencies and recurrent mutations in *CDH1* and *RHOA*. CIN tumours are, as the name suggests, highly chromosomal

unstable with focal amplifications of receptor tyrosine kinases, widespread demethylation patterns and frequent *TP53* mutations. MSI tumours are characterized by a high mutational burden, which is associated with a high number of somatic nucleotide polymorphisms, frequent INDELs and DNA hypermethylation or demethylation patterns. The most significant hypermethylated (and thus silenced) gene is *MLH1*, which is an essential component of the DNA repair pathway. Defects in the DNA repair pathway are believed to be responsible for the high mutation frequency within the coding region [13]. EBV-positive tumours are mainly found in the gastric body and fundus regions and are characterized by the presence of EBV DNA, frequent *PIK3CA* and *ARID1A* mutations and the overall highest DNA hypermethylation pattern, including an extreme CpG island methylation phenotype as shown for instance for the *CDKN2A* gene. Interestingly, the demethylation pattern is absent and *MLH1* is never epigenetically silenced in this subtype.

The gastric molecular classification can also be extended to GIACs [14], with the exception of EBV-positive tumours, which seem to be restricted to the stomach. MSI tumours are also found in the proximal colon, where PD-L1 is identified as a promising biomarker [16], which might be also applicable for gastric tumours. CIN tumours are found throughout the GI, but with some differences between the lower GI (LGI) and UGI tract, with regions of copy-number variations being focal in OAC and broader in the LGI. In CIN, *TP53* mutation alone is not sufficient for aneuploidy [17], but may facilitate the acquisition of secondary damage due to reactive oxygen species, gastric reflux or environmental signals that are different in various locations [14]. GS tumours are also found in both the UGI and LGI, but the acquired mutations are not overlapping. While *CDH1* and *RHOA* mutations are predominantly found in UGI, *KRAS* and *SOX9* are present in LGI.

The identification of the molecular subtypes prompted clinicians to correlate them to neoadjuvant response. There is a report showing that MSI-H GACs are non-responsive and may even progress upon standard chemotherapy [18], but these findings were not observed by others [19]. The lack of obvious correlations prevents the use of the molecular classification system as a clinical diagnostic tool. However, these genetic studies have revealed a series of potential druggable targets such as EZH2, BET and CDK4/6 for OACs [10] or PD-L1 for MSI gastric cancers [12]. However, most of them are not yet introduced in clinical practice for UGI tumours, and advances in *in vitro* cell culturing techniques might help to select the most promising candidates.

## 4. *In vitro* culture systems

Historically, human cancer-derived two-dimensional (2D) cell lines are the most widely used model for studying human tumour features, such as the cell line SKGT4 for oesophageal or AGS for gastric cancer. These cell lines are still frequently used as a model to test drug applicability [10,20], since they have fast proliferation rates, and are easy to handle and to genetically modify. However, these cell lines are generally derived from single cancerogenic subclones, which do not recapitulate the heterogeneity of original tumours [21]. In addition, extensive passaging led to the accumulation of genetic

**Table 1.** Establishment efficiency of PDOs from various cancer tissues [29,30,32–38].

| tumour type | success rate (%) |
| --- | --- |
| breast | 66 |
| colorectal | >90 |
| head and neck | 65 |
| oesophagus | 31 |
| ovarian | 85 |
| pancreas | 75–85 |
| prostate | 15–20 |
| stomach | 50 |

abnormalities that are not detected in the original tumour [22]. These factors complicate clinical translation of the findings.

In an attempt to overcome some of these limitations, patient-derived xenograft (PDX) models were developed, which has been recently reviewed [23]. In this model, human biopsies of resection specimens are transplanted into immunodeficient mice either heterotopically or orthotopically to keep the tissue alive and embedded in a more physiological environment [24]. This allows tumour vascularization and hypoxia to occur, which is otherwise not possible. Unfortunately, the availability of human tissue material, the long establishment time lasting several months and the requirement of large animal facilities prevent high-throughput drug screenings in PDX models [25,26].

In recent years, organoid culturing techniques have emerged, which allow the *in vitro* propagation and differentiation of adult organ-specific stem cells of healthy as well as tumour tissues [27–30]. These three-dimensional cultures can be expanded in short time and passaged over a long period of time, while they maintain features of the original epithelium in terms of overall architecture (e.g. lumen formation) and spontaneous cell differentiation processes [31]. Thereby, it is possible to study the homeostasis of the normal as well as the diseased state [9]. The efficiency of organoid establishment varies between tumours from different organs, as shown in table 1. If successful, little biological material is required to obtain enough cells for drug screens within a couple of weeks. The first published assay to screen a large array of drugs was performed on colorectal carcinoma PDOs [29]. In another study, PDOs were established from liver, pelvic, peritoneal and nodal metastasis of gastrointestinal cancer patients [39], and their mutational landscape was compared to the parental biopsies, revealing an overlap of 96%. Furthermore, good genotype–drug phenotype correlation was observed in drug screens. The proliferation of PDOs with specific gene amplification could be blocked by inhibiting the corresponding pathway. In both studies, initial counterintuitive findings were made. Among the colorectal PDOs, one organoid line lacking obvious *TP53* mutation was nonetheless resistant to Nutlin-3a. Closer inspection confirmed abnormal TP53 protein stabilization indicative of functional inactivation of the TP53 pathway via an unknown mechanism [29]. Similarly, one metastatic PDO line with EGFR amplification did not respond to the anti-EGFR inhibitor, as did the corresponding patient in the clinic [39]. These examples show that PDOs are

royalsocietypublishing.org/journal/rsob    Open Biol. **10**: 190274

royalsocietypublishing.org/journal/rsob    Open Biol. 10: 190274

superior to simple genotype–drug phenotype correlations. In the recent past, drug screens have also been reported for PDOs established from many other organs, including liver [40], ovarian [32] and stomach [33] tumours. In addition to drug screens, a recent study of head and neck tumours applied radiation screens in PDOs and compared the results with the clinical outcome [34]. Three patients relapsed after radiotherapy within one to six months, consistent with the finding that their organoid lines were classified as most resistant. Two other organoid lines were predicted to be good responders, and the corresponding patients also did not show any sign of relapse at the time of publication. These data suggest that PDOs can be used as good predictors for the efficiency of radiotherapy.

An important clinical feature of tumours is their clonal heterogeneity. While the majority of clones might respond to a given treatment, the survival of one is enough to generate overall resistance. It is thus crucial that PDOs support the growth of heterogeneous tumours, which was addressed by a study analysing the biopsies from multiple regions of three different colon tumours [41]. The established lines were subjected to whole-genome sequencing, which allowed to delineate the evolution trees of the tumours. PDOs from different regions of the same tumour harboured similar driver mutations, pointing towards early mutational events, as well as secondary mutations not found in adjacent tumour segments, indicative of later acquisition during tumour evolution. Maintenance of tumour-specific characteristics of tumour regions was also confirmed in other studies [33,39]. Overall, these experiments confirmed that organoid cultures support the growth of complex heterogeneous tumours, thus providing a great advantage over classical two-dimensional cell lines.

# 5. Progress with UGI organoids

Over the last year, there have been several efforts to develop oesophageal and gastric PDOs. The establishment of oesophageal organoid (OO) cultures is still problematic, as no long-term cultures of healthy adult OO have yet been reported. One study reported OO structures that are histologically comparable to the initial biopsies, but they could not be maintained long-term [42]. Of note, Trisno and colleagues [43] have been able to generate mature OO via a pluripotent stem cell differentiation protocol recapitulating oesophageal developmental steps. For adult OSCC organoids, six cultures were reported with an establishment efficiency of 43%. However, since these cultures were not compared with the original tissue, it is not clear how well they recapitulated the original tumorigenic features [44]. The most promising results were so far obtained for OAC, as 10 PDO lines could be established from resection specimen with an efficiency of 31% [37]. These organoids kept the identities of the original tumour tissue in terms of driver mutations and large-scale structural alterations and were genetically stable over a six-month culturing period. Differential drug sensitivities were observed but could not be correlated to patient response since the applied drugs were not yet in clinical use.

Research has progressed further on gastric epithelium, where healthy gastric organoids can be robustly established [45,46]. Three recent independent studies report the reliable generation of gastric tumour biobanks [33,36,47], although

with a slightly lower establishment efficiency than for healthy epithelium. Nanki and colleagues have generated 37 PDO lines and focused on the growth factor dependencies for culturing gastric tumour types. They obtained PDOs of all gastric tumour subtypes except for the EBV-associated ones. The established PDOs recapitulated the same histopathological features as the original tumour [39]. Yan and colleagues have performed the most thorough comparison of PDOs with their corresponding tumours. They have established 46 PDOs and classified them into the four gastric subtypes with comparable mutational spectra as previously defined by tumour sequencing studies. Interestingly, in both studies, the intestinal-type tumours grew as cohesive cystic organoids, the poorly differentiated tumours as solid structures and the diffuse-type ones as loosely cohesive cell clusters without any lumen [33,36]. Yan and colleagues also observed subclonal tumour evolution by comparing multiple biopsies obtained from primary tumours or even metastasis of the same patient. Initial drug screening data on nine PDOs suggested promising results for an ATR inhibitor administered to cancer cells with *ARID1A* mutations [33]. For another three patients, organoid data could be correlated to clinical patient response. Tumour metastasis of two patients decreased upon the administration of cisplatin and 5-FU, and the corresponding organoids seemed to be responsive as well. The third patient was resistant to capecitabine in the clinics as were the corresponding organoids in culture.

# 6. Clinical relevance and implementation

The therapeutic outcome of neoadjuvant and/or radiation treatment has been linked to tumour microenvironment in the clinics [48]. For example, a high stroma to tumour ratio is associated with a poor patient prognosis [49]. Additionally, immune cells play an important role in tumour clearance since dying cells free tumour-specific antigens, which are recognized by infiltrating cytotoxic T cells [50]. The relationship between immune cells and chemotherapy or radiation has been summarized elsewhere [50]. Here, we focus on the predictive value of organoid technology in clinical practice. In terms of microenvironment, organoids are simple systems since crucial growth factors, normally provided by the stroma, are added to the culture media. Nonetheless, original epithelial characteristics are kept *in vitro* over time and the absence of surrounding microenvironment even enables the characterization of pure tumour cell populations, which is otherwise hardly possible, as well as their specific response to chemotherapy or radiation. Efforts are under way to add back complexity to organoid cultures such as co-culturing them with immune cells [51]. While these experiments are important to characterize in greater detail the interplay between tumour and immune cells, it needs to be shown to what degree they will add to the predictive value of organoids in clinical set-ups. First trial experiments using organoids for radiation and drug screens showed good correlation with the respective *in vivo* patient responses [33,34].

Overall, organoid cultures hold the great promise to predict the individual response to a wide range of drugs or radiation, as shown by studies that correlated the response to certain genetic mutations by a genotype-drug phenotype association. However, in clinical practice, the genetic make-up of the tumour, which could be informative for an *a priori*

selection of a tailored treatment, is not available for most patients at the start of the clinical treatment regimen. Therefore, a different approach could be to predict the response efficiency of patients to already known clinical treatment regimens such as CROSS or FLOT prior to their administration. Such an approach would allow screening for the best possible treatment regimen or to identify non-responders that would benefit from direct surgery, thereby omitting any neoadjuvant or radiation treatment, which is often accompanied with severe side effects and potential tumour progression. The correlation between clinical and organoid data of oesophageal and gastric cancer is up to now only anecdotal but provides already encouraging results. While larger-cohort studies are required to confirm the accuracy of the predictive value of organoid drug response, two additional important points have to be addressed before clinical application. First, published oesophageal and gastric data are mainly derived from surgical resection specimen obtained after preoperative treatment that reduces the amount of viable tumour cells and/or may prevent organoid outgrowth, especially in the case of (near-)complete responders. This results in lower organoid establishment efficiencies, which is most likely to be responsible for the 31% rate of OAC outgrowth [37]. It is, however, expected that this improves if neoadjuvant naïve cells from pre-treatment biopsies are used. Alternatively, different culture media have been established for oesophageal tumour organoids (OAC [37] and OSCC [44]), and some OAC lines that are not capable of proliferating in OAC media may grow in OSCC media, and vice versa. Additionally, molecular tumour subtypes may have different growth factor dependencies, which has not yet been analysed in detail. It is thus advisable to initiate cultures in multiple well-characterized culture media such as healthy oesophageal, OAC, OSCC, gastric or small intestinal media [28,37,42,44,45] to improve the overall establishment rate. Second, is it possible to obtain organoid data within a short time frame that can be implemented within the clinical diagnostic process? On average, it takes three weeks from diagnosis to the start of treatment. In this short clinical time frame, fast-growing organoid lines can be screened, but the observed substantial heterogeneity in the growth rate of different PDO lines remains a challenge.

While these challenging aspects need to be addressed, the overall outlook for the use of organoids in predicting clinical outcome is promising.

## 7. Discussion

Current multicentre randomized trials in UGI cancers focus on one-size-fits-all treatment strategies with relatively poor overall response rates [6]. This is explained by the large genetic diversity between the different subtypes and their subclones. It is therefore expected to be more effective to switch to a personalized approach. As discussed in this review, current data suggest a good predictive value of PDOs in drug and radiation assays, even if the overall mutational landscape is unknown. In fact, the functional readout is even superior to simple genotype–drug phenotype correlations [39]. Once organoid experiments can reliably predict the individual response of each patient to existing treatment regimens, such as CROSS and FLOT, they will allow the selection of the most promising treatment strategy. The feasibility and accuracy of this approach needs, however, to be confirmed by studying a larger number of cases, whose patient response is correlated with the organoid response. In The Netherlands, two ongoing trials, TUMOROID (NL49002.031.14) and OPTIC (NL61668.041.17), are comparing the predictive value of the organoid treatment response to the clinical outcome of the corresponding patients with metastatic colon, breast or non-small cell lung cancers, or of first-line metastatic colorectal cancer patients who did not receive any treatment before. Similar trials will be required for UGI cancers, for which only anecdotal data with a promising trend exist so far.

Data accessibility. This article has no additional data.

Authors' contributions. G.A.B., M.F.G.d.M. and H.C. designed the outline of this review; G.A.B. wrote the molecular and organoid part with the help of I.A.F. and H.C.; F.L. and M.F.G.d.M. summarized the clinical aspects of this review with the help of R.v.H. and J.P.R.

Competing interests. H.C. is inventor on several patents related to organoid technology; his full disclosure is given at https://www.uu.nl/staff/JCClevers/.

Funding. This work was supported by ZonMw 114021012 and Oncode.

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
