## [Reviewer comments · Open Biology]

Review History

RSOB-19-0274.R0 (Original submission)

Review form: Reviewer 1

Recommendation

Major revision is needed (please make suggestions in comments)

Do you have any ethical concerns with this paper?

No

Comments to the Author

This paper provides a timely and useful summary of upper GI malignancy, covering current clinical practice and issues/dilemmas surrounding how best to establish optimal therapies for patients, given the huge heterogeneity within and between patients at a molecular level. It goes on to describe progress to date with establishing in vivo and in vitro models of upper GI malignancy, setting it in the context of progress with other cancer types, and discussing how such models might be integrated into clinical trials to allow personalisation of medicine.

Overall, this is a thorough review and useful for researchers in the field to help focus areas that

need improvement. However, although the low take-rates for UGI organoids is mentioned, the challenge this poses is not sufficiently brought out in the discussion. Since the study highlighted as the most successful from this point of view had a success rate of only 31%, improving this would be a major hurdle for carrying out trials since organoids would need to be generated for all patients. Some discussion of how such take-rates might be improved in the future would have been valuable. Furthermore, there is no discussion of the potential impact of aspects of the tumour microenvironment not represented in organoids, such as that arising from interaction with immune and mesenchymal cells which are known to provide signals that influence many aspects of cancer cell biology including drug sensitivity.

Decision letter (RSOB-19-0274.R0)

05-Feb-2020

Dear Dr Busslinger,

We are writing to inform you that the Editor has reached a decision on your manuscript RSOB-19-0274 entitled "The potential and challenges of patient-derived organoids in guiding the multimodality treatment of upper gastrointestinal malignancies", submitted to Open Biology.

As you will see from the reviewer's comments below, there are a number of criticisms that prevent us from accepting your manuscript at this stage. The reviewer suggests, however, that a revised version could be acceptable, if you are able to address their concerns. If you think that you can deal satisfactorily with the reviewer's suggestions, we would be pleased to consider a revised manuscript.

The revision will be re-reviewed, where possible, by the original referees. As such, please submit the revised version of your manuscript within four weeks. If you do not think you will be able to meet this date please let us know immediately.

When submitting your revised manuscript, please respond to the comments made by the referee(s). In order to expedite the processing of the revised manuscript, please be as specific as possible in your response to the referee(s).

Please see our detailed instructions for revision requirements
<https://royalsociety.org/journals/authors/author-guidelines/>

Sincerely,

The Open Biology Team
mailto: openbiology@royalsociety.org

Reviewer's Comments to Author(s):

This paper provides a timely and useful summary of upper GI malignancy, covering current clinical practice and issues/dilemmas surrounding how best to establish optimal therapies for patients, given the huge heterogeneity within and between patients at a molecular level. It goes on to describe progress to date with establishing in vivo and in vitro models of upper GI malignancy, setting it in the context of progress with other cancer types, and discussing how such models might be integrated into clinical trials to allow personalisation of medicine.

Overall, this is a thorough review and useful for researchers in the field to help focus areas that need improvement. However, although the low take-rates for UGI organoids is mentioned, the challenge this poses is not sufficiently brought out in the discussion. Since the study highlighted as the most successful from this point of view had a success rate of only 31%, improving this would be a major hurdle for carrying out trials since organoids would need to be generated for all patients. Some discussion of how such take-rates might be improved in the future would have been valuable.

Furthermore, there is no discussion of the potential impact of aspects of the tumour microenvironment not represented in organoids, such as that arising from interaction with immune and mesenchymal cells which are known to provide signals that influence many aspects of cancer cell biology including drug sensitivity.

RSOB-19-0274.R1 (Revision)

Review form: Reviewer 1

Recommendation

Accept as is

Do you have any ethical concerns with this paper?

No

Comments to the Author

The authors have addressed the referee's comments appropriately

Decision letter (RSOB-19-0274.R1)

19-Mar-2020

Dear Dr Busslinger

We are pleased to inform you that your manuscript entitled "The potential and challenges of patient-derived organoids in guiding the multimodality treatment of upper gastrointestinal malignancies" has been accepted by the Editor for publication in Open Biology.

You can expect to receive a proof of your article from our Production office in due course, please

check your spam filter if you do not receive it within the next 10 working days. Please let us know if you are likely to be away from e-mail contact during this time.

Sincerely,

The Open Biology Team
mailto: openbiology@royalsociety.org